# Optimal n-Type Al-Doped ZnO Overlayers for Charge Transport Enhancement in p-Type Cu_2_O Photocathodes

**DOI:** 10.3390/mi12030338

**Published:** 2021-03-22

**Authors:** Hak Hyeon Lee, Dong Su Kim, Ji Hoon Choi, Young Been Kim, Sung Hyeon Jung, Swagotom Sarker, Nishad G. Deshpande, Hee Won Suh, Hyung Koun Cho

**Affiliations:** 1School of Advanced Materials Science and Engineering, Sungkyunkwan University, 2066 Seobu-ro, Jangan-gu, Suwon, Gyeonggi-do 16419, Korea; zadxs@skku.edu (H.H.L.); dskim2846@naver.com (D.S.K.); Jihoon9290@skku.edu (J.H.C.); yb3201@skku.edu (Y.B.K.); wjdtjdgus2@skku.edu (S.H.J.); naekkeo@skku.edu (H.W.S.); 2Research Center for Advanced Materials Technology, Sungkyunkwan University, 2066 Seobu-ro, Jangan-gu, Suwon, Gyeonggi-do 16419, Korea; swagotom@yahoo.com (S.S.); nicedeshpande@gmail.com (N.G.D.)

**Keywords:** photoelectrochemical, Cu_2_O, photocathodes, Al-doped ZnO, charge transport, ALD

## Abstract

An effective strategy for improving the charge transport efficiency of p-type Cu_2_O photocathodes is the use of counter n-type semiconductors with a proper band alignment, preferably using Al-doped ZnO (AZO). Atomic layer deposition (ALD)-prepared AZO films show an increase in the built-in potential at the Cu_2_O/AZO interface as well as an excellent conformal coating with a thin thickness on irregular Cu_2_O. Considering the thin thickness of the AZO overlayers, it is expected that the composition of the Al and the layer stacking sequence in the ALD process will significantly influence the charge transport behavior and the photoelectrochemical (PEC) performance. We designed various stacking orders of AZO overlayers where the stacking layers consisted of Al_2_O_3_ (or Al) and ZnO using the atomically controlled ALD process. Al doping in ZnO results in a wide bandgap and does not degrade the absorption efficiency of Cu_2_O. The best PEC performance was obtained for the sample with an AZO overlayer containing conductive Al layers in the bottom and top regions. The Cu_2_O/AZO/TiO_2_/Pt photoelectrode with this overlayer exhibits an open circuit potential of 0.63 V and maintains a high cathodic photocurrent value of approximately −3.2 mA cm^−2^ at 0 V_RHE_ for over 100 min.

## 1. Introduction

Among the various technologies for the sustainable production of hydrogen, photoelectrochemical (PEC) water splitting using photoactive materials is a promising and low-cost method with almost zero carbon emissions [1,2]. Pioneered by the work on n-type TiO_2_ [3,4,5], numerous studies have been conducted on the use of various n-type oxide photoanodes such as WO_3_ [5,6,7], BiVO_4_ [7,8], and Fe_2_O_3_ [5,9,10]. However, research on the photocathodes, a latecomer, is lacking. It is therefore necessary to emphasize the importance of photocathodes that can generate direct hydrogen products and to conduct extensive research in this field. Among various p-type photocathodes, copper-based oxide materials have received the most attention. Cu_2_O is one of the most promising p-type semiconductors for reducing water to hydrogen owing to the suitable conduction band position and its good visible absorption [11,12]. With a bandgap of approximately 2 eV, Cu_2_O can efficiently absorb visible light. Theoretically, Cu_2_O can generate a photocurrent of 14.7 mA cm^−2^ under AM (air mass) 1.5G illumination (100 mW cm^−2^), which corresponds to a maximum solar-to-hydrogen (STH) conversion efficiency of 18% [13]. Moreover, copper is a nontoxic and earth-abundant material, and Cu_2_O can be easily produced by low-temperature and nonvacuum processes such as electrodeposition [14,15].

The photocurrent and energy conversion efficiencies of Cu_2_O in PEC water splitting suffer from some limitations that need to be resolved. 

The Cu_2_O absorption layer forms a sufficient number of electron-hole pairs (EHPs) but lacks the driving force to move minority carriers adequately because of its low carrier mobility. In other words, the efficiency of the charge separation and transport of photogenerated electrons was low; it is challenging to sufficiently utilize the photogenerated electrons for hydrogen generation reactions with only a single Cu_2_O layer.Self-photocorrosion occurs when the electrolyte and Cu_2_O surface are in direct contact. As a result, long-term stability cannot be ensured owing to the phase change and collapse of the absorber materials [16,17].

The formation of a p-n junction may be an effective method to overcome these weaknesses and improve the performance of Cu_2_O photocathodes. The structure of the absorbent layer and an appropriate heterojunction can generate additional built-in potential and increase the photovoltage as well as the photocurrent by overcoming the insufficient charge separation and transfer efficiencies [11,18,19,20]. n-Type ZnO has a transmittance of more than 80% owing to its wide bandgap and has a suitable band alignment with Cu_2_O [21,22,23,24]. Thus, n-type ZnO does not cause an absorption loss to Cu_2_O during light irradiation. The transferred charges must survive without loss in the ZnO bulk. Therefore, it is expected that the efficiency can be further improved by using an efficient p-Cu_2_O/n-ZnO heterojunction photocathode fabricated via doping, which will have an increased conductivity of ZnO. In general, undoped ZnO has an insufficient electrical conductivity for application as a transparent conductive oxide; therefore, group III elements such as Al, Ga, and In are added as electron donors to ZnO. Al-Doped ZnO (AZO) is a good candidate with a high carrier concentration and excellent electrical conductivity [11]. AZO is also compatible with low-temperature processes and has stable properties under thin-film deposition, thereby making it suitable for industrial manufacture. AZO thin films can be deposited by the sol-gel process [25], chemical vapor deposition [26], magnetron sputtering [27], pulsed laser deposition [28], and atomic layer deposition (ALD) [11]. Among them, the magnetron sputtering process is generally used for AZO thin film deposition. However, it has the drawback of requiring a subsequent annealing treatment process. The ALD method is based on a self-limiting reaction at a relatively low temperature; the reaction is termed self-limiting because reactions occur only between the reactant and the surface. When the surface changes to a homogenous atomic monolayer, no further reaction occurs, even if an excessive amount of reactive gas is supplied [29]. Therefore, the thickness of the film can be easily controlled by adjusting the number of cycles. ALD is attracting increasing attention as an important deposition technology for the production of nanoscale transparent conductive oxide (TCO) films because it allows the deposition of thin films with excellent uniformity on various substrates. Although numerous ALD-based studies on Al-doped Zn-base oxides have been conducted [13,30,31], not many studies have focused on the effect of ZnO-based overlayers on the photoelectrochemical reaction.

In this study, we designed and fabricated p-Cu_2_O/n-ZnO:Al (AZO) heterojunctions with various overlayers as photocathodes for PEC cells. Herein, 20-nm-thick ZnO-based layers were doped with Al dopants under various conditions on the Cu_2_O thin films by thermal ALD. The p-Cu_2_O/n-AZO heterojunction photocathodes showed an excellent PEC performance for water splitting with an enhanced photocurrent of −3.2 mA cm^−2^ at 0 V vs. RHE under AM 1.5G illumination. The formation of a suitable heterojunction at the interface between the light-absorbing layer and the overlayer successfully increased the charge transport capacity.

## 2. Materials and Methods

### 2.1. Synthesis of Cu_2_O Photocathode

For Cu_2_O electrodeposition, ITO with a sheet resistance of 10 Ω sq^−1^ deposited on glass was used as the working electrode. Before electrodeposition, the ITO glass substrates were cut into sheets of 2 cm × 3 cm in size and then washed by sonication in acetone, ethanol, and deionized water for 20 min each. A portion of the exposed ITO glass substrate was masked with Kapton tape so that only a 1 cm × 1 cm area was left exposed. The electrodeposition electrolyte is a lactate-stabilized copper solution with 0.4 M of copper sulfate (CuSO_4_, >98%, Junsei, Kyoto, Japan) and 3 M of lactic acid (85% solution, Sigma-Aldrich, St. Louis, MO, USA). The pH was adjusted to 11 using 4 M sodium hydroxide (NaOH, >98%, Sigma-Aldrich, Stockholm, Sweden) solution. Electrodeposition was carried out potentiostatically at −0.5 V using a 3-electrode system (VersaSTAT4 Potentiostat, AMETEK Princeton Applied Research, Oak Ridge, TN, USA) with a Pt mesh as the counter electrode and an Ag/AgCl (saturated KCl) reference electrode. The thickness of the electrodeposited Cu_2_O layer was controlled using the chronocoulometry method to obtain a 2-μm-thick layer.

### 2.2. Preparation of Cu_2_O/AZO Heterojunction

ZnO and Al-doped ZnO thin layers were deposited on Cu_2_O films using an ALD system (Lucida D100, NCD, Daejeon, Korea). The overlayers were deposited at a chamber temperature of 423 K using diethylzinc (DEZn) as a zinc precursor, trimethylaluminum (TMA) as an aluminum precursor, and deionized water at a precursor temperature of 283 K. The deposition was carried out under an N_2_ flow of 100 sccm with a pulse length of 0.2 s for the metal precursor, 0.1 s for deionized water, and a pause period of 10 s. For the deposition of the 20-nm-thick ZnO layer, 100 cycles of the DEZn step were carried out, and one cycle of the TMA step was inserted between every 20 cycles of the DEZn step for the AZO deposition.

### 2.3. Synthesis of TiO_2_ Passivation Layers and Pt Catalysts

The TiO_2_ passivation layer was deposited on the Cu_2_O/(ZnO or AZO) photoelectrodes using an ALD system (iOV m100, iSAC research, Daejeon, Korea) at 423 K. Tetrakis dimethylamino titanium (TDMAT) was used as the Ti precursor together with deionized water. Pt, used as an HER catalyst, was galvanostatically photoelectrochemically deposited on the surface of TiO_2_ at a current density of −8.5 μA cm^−2^ for 15 min in a solution containing 1 mM H_2_PtCl_6_ under AM 1.5G illumination.

### 2.4. Sample Characterization

The morphological and crystalline properties were characterized by field-emission scanning electron microscopy (FE-SEM, JSM-6700F, 10 kV) and X-ray diffraction (XRD, Bruker AXS D8 Discover, USA, Cu Kα radiation source). The optical absorbance was analyzed using a UV-Vis-NIR spectrophotometer (Cary 5000, Agilent, Santa Clara, CA, USA). The electrical properties were measured using In electrodes by an I-V station (Keithley 4200A-SCS, Tektronix, Beaverton, OR, USA) and 4-point probe system (CMT-SR2000N, AIT, Suwon, Korea).

### 2.5. PEC Measurements

The photoelectrochemical performance of the prepared Cu_2_O-based photoelectrodes was evaluated using a 3-electrode system under AM 1.5G illumination (100 mW cm^−2^). The light source is a 150 W xenon lamp with an AM 1.5G filter. The electrolytes contained 1 M sodium sulfate (Na_2_SO_4_ > 99%, Sigma-Aldrich, Bengaluru, India) solution buffered with 0.2 M potassium phosphate (KH_2_PO_4_ > 99%, Sigma-Aldrich, Tokyo, Japan) and boric acid (H_3_BO_3_ > 99%, Sigma-Aldrich, India) to adjust the pH to 5. Electrochemical impedance spectroscopy (EIS) was performed in the frequency range of 1 MHz to 1 Hz at 0 V_RHE_, and Mott–Schottky plots were obtained at 1000 Hz and 10 mV to estimate the carrier concentration and flat band potential.

## 3. Results and Discussion

### 3.1. Characterization of Cu_2_O/ZnO and Cu_2_O/AZO Structures

To maximize the PEC performance of the p-type Cu_2_O photocathodes after the absorption of irradiated light, typical p-n heterojunctions with n-type overlayers can be employed. There are three key factors to be considered: (1) Ensuring the generation of EHPs with no degradation of the optical absorption efficiency of p-type Cu_2_O. This requires the use of transparent overlayers with wider bandgaps than those in Cu_2_O. (2) An adequate band alignment favored by the charge separation and charge transport behaviors in the heterojunction. (3) Suppressing the recombination loss of the charges transported from the Cu_2_O in the overlayers. These factors are strongly related to the electrical performance of n-type overlayers.

We selected general n-type ZnO-based overlayers (ZnO and AZO) and deposited them with a thickness of 20 nm using ALD. The details of the used overlayers are listed in Table 1. The typical morphology and phase of the Cu_2_O/ZnO heterojunction photocathodes prepared on ITO substrates were characterized by SEM, XRD, and UV-Vis-NIR absorbance spectra. Figure 1a shows the top view of the morphology of the Cu_2_O microcrystalline film after coating with 20-nm ZnO deposited at 423 K. Higher deposition temperatures induce the phase change of Cu_2_O into CuO, which severely degrades the PEC performance [20]. Because the ALD coating is typically homogeneous and there is no phase change at the used temperature, no apparent changes in the morphology of the microcrystals were detected in the samples before and after coating. As shown in the cross-sectional images in Figure 1b,c, the average thickness of the ZnO overlayers was approximately 20 nm, and the entire surface of the Cu_2_O was fully covered with ZnO, resulting in the prevention of direct contact between the photocathode and electrolyte. This leads to the effective suppression of photocorrosion on Cu_2_O and the formation of a uniform p-n junction. The XRD measurements revealed only Cu_2_O diffraction peaks in the Cu_2_O/ZnO sample (Figure 1d), except for ITO peaks. The absence of ZnO signals may be due to the small thickness and low crystallinity of the ZnO layers. It is necessary to accurately analyze the optical properties in order to check whether the ~20-nm-thick overlayers, formed on the surface of the absorption layer, affect the light absorption of the Cu_2_O layer.

It has been reported that doping with Al increases the bulk conductivity of the ZnO-based overlayers grown by ALD [32]. However, the stacking sequence of Al doping, such as the number and position, can significantly influence the electrical conductivity owing to the thin overlayer thickness. Nevertheless, detailed studies on the variation of the PEC performance of the photocathodes with respect to the Al doping stacking order are lacking. Figure 2 shows the visible-light absorbance of Cu_2_O, the overlayers, and the Cu_2_O/overlayer samples. The absorbance is converted from the transmittance of each sample by the Beer–Lambert Law, as shown in Equation (1) [33]:A = −log(T)(1)
where A is the absorbance and T is the transmittance. Additionally, the absorption coefficient α is calculated by the following Equation (2):α = 2.303 A/d(2)
where d is the film thickness.

The bandgaps can be estimated from Tauc’s plots using the absorption coefficient and transmittance, as shown in Equation (3) [34]:α (hν) = C(hν−E_g_)^1/2^(3)
where α is the optical absorption coefficient, hν is the photon energy of the incident light, C is a constant, and E_g_ is the optical bandgap energy. The direct bandgap of the developed Cu_2_O was estimated to be 2.41 eV from the Tauc’s plot. The overlayers of the ZnO, AZ@AZO, AA@AZO, ZA@AZO, and ZZ@AZO thin films have bandgaps between 3.22 and 3.33 eV (Figure 2b, Appendix A). The higher Al doping in the AZO films via the insertion of an additional Al layer induces a slight increase in the bandgap. This increase is ascribed to the Burstein–Moss effect, in which the increase in free-electron density causes an increased optical bandgap [28]. Fortunately, the very thin protective n-type overlayers on the surface of the electrodeposited Cu_2_O film did not induce any significant changes in the bandgap or absorbance spectra of the protected photoelectrodes (Figure 2c) compared to that of the pristine electrodeposited Cu_2_O. This implies that the use of these overlayers does not hinder the optical absorption of the photocathode Cu_2_O films, even though satisfactory p-n junctions were formed.

### 3.2. Electrical Characteristics of ZnO-Based Overlayers

The electron charges passing through the p-n heterojunction react with the electrolytes to produce hydrogen. Thus, it is necessary to track the path of the carriers in order to improve the photoelectrical reaction. The electrons transported along the conduction band of the Cu_2_O/overlayer heterojunction with an adequate band alignment can undergo recombination within the overlayers. This process is strongly related to the electrical conductivity and mobility of overlayers. Because the nominally undoped ZnO has a low electrical conductivity, it is expected to show a considerable loss in carrier transport. The selection of the first layers as well as the stacking sequence of the Al added as the electron donor via ALD should be investigated in detail for an optimal PEC operation. Thus, various ZnO-based overlayers were prepared with the same thickness (Table 1). Here, the AA@AZO and AZ@AZO samples utilize injected Al precursors as the first layer of the AZO coating, while the last layer for the AA@AZO and ZA@AZO samples is an Al monolayer. The electrical conductivities of the overlayers were compared with those of In-(Al)ZnO-ITO with a vertical stacking configuration (Figure 3a). All samples with Al doping exhibited an improved electrical conductivity, regardless of the stacking sequence. The lowest resistivity was obtained from the AA@AZO overlayer with Al monolayers in the bottom and top regions (Figure 3b). The AZ@AZO and ZA@AZO samples with one Al monolayer also showed better conductivity than the ZZ@AZO sample without any Al monolayer. This proves the effectiveness of using an Al monolayer in the bottom or top layers to improve the electrical conductivity.

### 3.3. Visible-Light PEC Behavior of Samples with Different Overlayers

To confirm the improvement in the charge collection with the use of various overlayers, an electrochemical impedance spectroscopy (EIS) study was performed at 0 V vs. RHE to investigate the impedance at the Cu_2_O/overlayers’ interfaces under AM 1.5G illumination. The EIS curves are displayed in the form of a Nyquist plot in Figure 4, and an equivalent circuit is presented in Appendix A. In the equivalent circuit, R_1_ is the series resistance (including the resistance between ITO and Cu_2_O), R_2_ is the resistance of the Cu_2_O bulk and the charge transport through the overlayers, and R_3_ represents the resistance for the charge transfer at the interface with the electrolyte. C_1_ is the capacitance of the space charge region at the Cu_2_O/overlayer interfaces, and C_2_ is the surface space charge capacitance of the overlayers. C_3_ in the outermost shell is the capacitance of the electric double layer. Figure 4b shows that the photocathode with AA@AZO exhibited the lowest R_1_ among the prepared samples. The reduced C_1_ in Cu_2_O indicates that photocarriers generated in Cu_2_O were transported to the AA@AZO layers faster than in the other samples. Because AZ@AZO-overlayered Cu_2_O showed a similarly low R_1_ as AA@AZO-overlayered Cu_2_O, we found that the optimal p-n junction in the Cu_2_O surface was formed using AZO with an Al coating as the first monolayer. In addition, the lower R_2_ of AA@AZO-overlayered Cu_2_O implies that the AA@AZO layer has a higher conductivity than the other overlayers, which is in accordance with the results shown in Figure 3. Therefore, by depositing an AA@AZO overlayer on the surface of Cu_2_O, the transport efficiency can be considerably enhanced.

To investigate the reason behind the enhanced transport efficiency of AA@AZO-overlayered Cu_2_O, the carrier density and band structures were analyzed, and the results are shown in Figure 5. The photoactive Cu_2_O/AA@AZO electrodes were characterized by a Mott–Schottky analysis to evaluate the flat band potential (V_fb_) and carrier density (N_A_) for band structure alignment. The Mott-Schottky plots of the ZnO and AZO single layers were analyzed using the following equation [35]:C^−2^ = 2(eεε_0_N_A_A^2^)^−1^(V − V_fb_k_B_Te^−1^)(4)
where C is the space charge capacitance, ε_0_ the permittivity of free space, ε the dielectric constant of ZnO and AZO (taken as 3.8) [36], e the elementary charge, A the electrode area, V the applied potential, T the temperature, and k_B_ the Boltzmann constant. Compared with pure ZnO, there is an increase in the carrier density and a negative shift in the flat band potential in AZO overlayers (Figure 5a). As shown in Appendix A, the AA@AZO overlayer shows a superior V_fb_ of −0.282 V vs. RHE and a high carrier density of 1.58 × 10^18^ cm^−3^. This doping behavior is demonstrated schematically in the band diagrams in Figure 5b,c. The use of the AA@AZO overlayer induces a relatively abrupt band bending at the Cu_2_O/AZO interface, which is expected to enhance charge transport because of the stronger built-in potential. The band structure of Cu_2_O has been described in a previous study [37].

### 3.4. Enhancement in Charge Transfer and PEC Activity

To evaluate the photoelectrochemical performance of the Cu_2_O/overlayer photoelectrodes, linear sweep voltammetry (LSV) was conducted under AM 1.5G illumination. As shown in Figure 6a, the Cu_2_O/AZO electrodes show a better performance than the Cu_2_O/ZnO electrode with a current of > 1.6 mA cm^−2^ at 0 V_RHE_. Among various overlayers, the Cu_2_O/AA@AZO electrode exhibited the best open circuit potential (OCP) of 0.63 V_RHE_ and a current density of –2.9 mA cm^2^ at 0 V_RHE_. This is consistent with the EIS and Mott-Schottky analyses. The AA@AZO layer has the lowest transport resistance among the samples and can induce a higher photovoltage and a lower flat band potential through the junction with Cu_2_O. In addition, Cu_2_O/AA@AZO exhibits a better fill factor than the other samples. To test the long-term stability of the Cu_2_O/AA@AZO electrode, a 20-nm TiO_2_ passivation layer and a Pt catalyst, respectively prepared by ALD and photoelectrochemical deposition, were deposited on the surface of AZO. From the chronoamperometry measurement at 0 V_RHE_, it can be seen that the Cu_2_O/AA@AZO/TiO_2_/Pt photoelectrode maintains a high cathodic photocurrent value of about −3.2 mA cm^−2^ for over 100 min.

## 4. Conclusions

We designed optimal AZO overlayers for Cu_2_O photocathodes with an enhanced charge transport and evaluated the properties of the AZO layers. The transparent ZnO and AZO, which have a wide bandgap of over 3.22 eV, suppressed the absorption of incident photons before they arrived at the Cu_2_O absorbing layers. The AZO layer has an increased bandgap (3.33 eV) due to the Al doping effect. By controlling the steps for Al injection during ALD, the AZO layers could be made to exhibit a higher electrical conductivity with increased Al doping. The Cu_2_O/AZO heterojunction generated a higher photovoltage via Al injection in the bottom and top regions of the AZO because of the enhanced built-in potential. The Cu_2_O/AA@AZO photoelectrode exhibited an optimal photocurrent of −3.2 mA cm^−2^ with TiO_2_ passivation and a Pt catalyst under AM 1.5G illumination.

## Figures and Tables

**Figure 1 micromachines-12-00338-f001:**
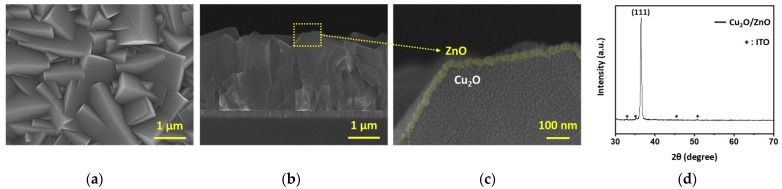
(**a**) Top view, (**b**,**c**) cross-view SEM images, and (**d**) X-ray diffraction pattern of the ZnO-overlayered Cu_2_O film.

**Figure 2 micromachines-12-00338-f002:**
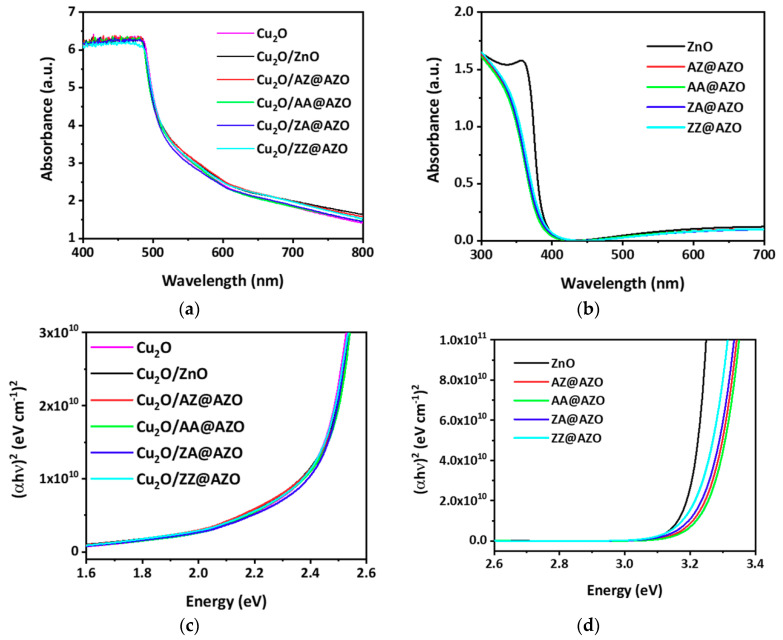
(**a**,**b**) UV-Vis-NIR absorption spectra, and (**c**,**d**) Tauc plots of (**a**,**c**) pristine and overlayered Cu_2_O and (**b**,**d**) ZnO and AZO overlayers.

**Figure 3 micromachines-12-00338-f003:**
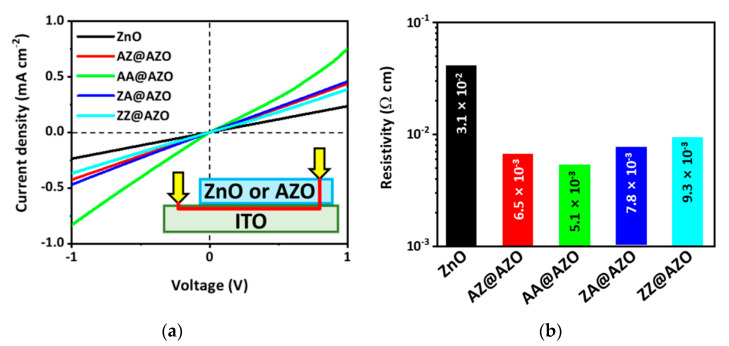
(**a**) I-V graphs of ZnO and AZO overlayers on ITO substrates; (**b**) Resistivity values from the 4-point probe method of 20-nm-thick ZnO and AZO thin films on glass substrates.

**Figure 4 micromachines-12-00338-f004:**
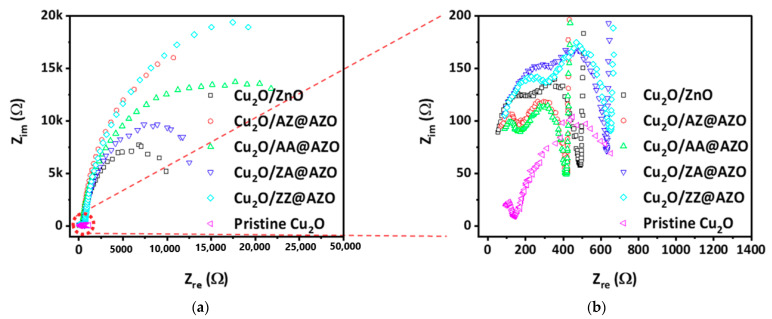
(**a**,**b**) Electrochemical impedance spectroscopy results of pristine Cu_2_O and Cu_2_O/overlayer electrodes under AM 1.5G illumination.

**Figure 5 micromachines-12-00338-f005:**
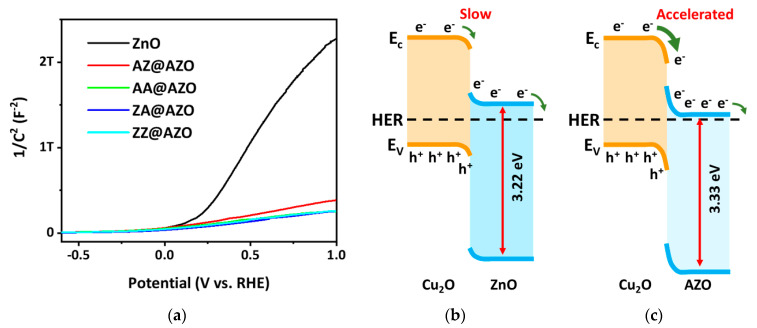
(**a**) Mott–Schottky plots of ZnO and AZO overlayers. Schematic band diagrams of the (**b**) Cu_2_O/ZnO and (**c**) Cu_2_O/AA@AZO junctions estimated from the Mott-Schottky plots.

**Figure 6 micromachines-12-00338-f006:**
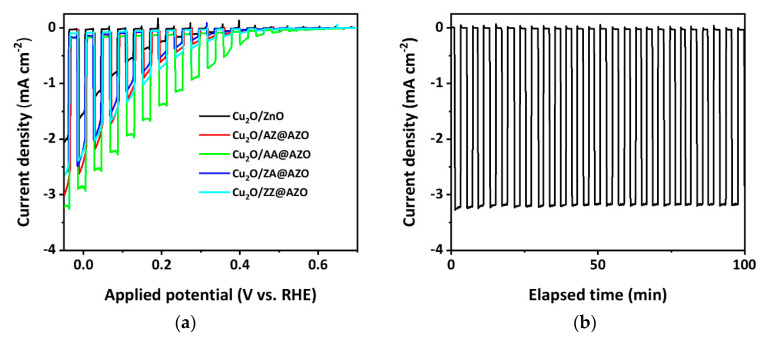
(**a**) Linear sweep voltammetry properties of prepared Cu_2_O/overlayer photocathodes; (**b**) Long-term stability test of Cu_2_O/AA@AZO/TiO_2_/Pt photoelectrode at 0 V_RHE_ in a 1 M Na_2_SO_4_ solution buffered with KH_2_PO_4_ and H_3_BO_3_ under AM 1.5G illumination.

**Table 1 micromachines-12-00338-t001:** Nomenclature used for the prepared samples.

Sample (Cycle)	Name
ZnO (20) × 5: 20 nm	ZnO
(Al (1) + ZnO (20)) × 5 (ZnO end): 20 nm	AZ@AZO
(Al (1) + ZnO (20)) × 5 + Al (1) (Al end): 20 nm	AA@AZO
(ZnO (20) + Al (1)) × 5 (Al end): 20 nm	ZA@AZO
(ZnO (20) + Al (1)) × 4 + (ZnO end): 20 nm	ZZ@AZO

## Data Availability

Data available upon request from the corresponding author.

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
