# Peer review of "Optimal n-Type Al-Doped ZnO Overlayers for Charge Transport Enhancement in p-Type Cu2O Photocathodes"

_micromachines, 2021, doi:10.3390/mi12030338_

Round 1
Reviewer 1 Report
See the attached document.

Reviewer 2 Report
The manuscript “Optimal n-type Al-doped ZnO overlayers for charge transport enhancement in p-type Cu2O photocathodes “ by : Hak Hyeon Lee, Dong Su Kim, Ji Hoon Choi, Young Been Kim, Sung Hyeon Jung, Swagotom Sarker, Nishad G. Deshpande , Hee Won Suh and Hyung Koun Cho is devoted to study the influence of Al-doped ZnO overlayers on photoelectrochemical performance of Cu2O photocatodes. Authors clamed that the Cu2O/AZO electrodes show better performance than the Cu2O/ZnO electrode with a current of > 1.6 mA cm-2 at 0.
There are some mistakes:
1) It is known that ZnO and Al-doped ZnO thin layers growth by ALD system has a good conductivity. However the high resistivity of layers depicted in Figure 3(b) are not real.
2) In Figures 2(c) and 2(d) the values of (αհν) are not correct. Also in Figure 2 (a) and (b) the absorbance is needed to show in real unit for credibility.
Therefore, major revision is required for this manuscript before publication.
Round 2
Reviewer 2 Report
The manuscript “Optimal n-type Al-doped ZnO overlayers for charge transport enhancement in p-type Cu2O photocathodes “ by : Hak Hyeon Lee, Dong Su Kim, Ji Hoon Choi, Young Been Kim, Sung Hyeon Jung, Swagotom Sarker, Nishad G. Deshpande , Hee Won Suh and Hyung Koun Cho has been slightly improved. However, it will be noted that the values of (αհν) (see Figures 2(d)) are different from known in the literature (such as, Tai Nguen et al. Results in materials 6 (2020) 100088). A discussion is highly desirable. The authors don´t take into account my remark „in Figure 2 (a) and (b) the absorbance is needed to show in real unit for credibility”
Thereby, I think the manuscript can be published in Micromachines after a revision.
